# Pool-Based Genetic Programming Using Evospace, Local Search and Bloat Control

**Perla Juárez-Smith** [1], **Leonardo Trujillo** [1,*] , **Mario García-Valdez** [1],
**Francisco Fernández de Vega** [2] **and Francisco Chávez** [3]

[1] Tecnológico Nacional de México/Instituto Tecnológico de Tijuana, Tijuana BC C.P. 22430, Mexico
[2] Departamento de Tecnología de los Computadores y de las Comunicaciones, Universidad de Extremadura, 06800 Mérida, Spain
[3] Departamento de Ingeniería Sistemas Informáticos y Telemáticos, Universidad de Extremadura, 06800 Mérida, Spain
* Correspondence: leonardo.trujillo@tectijuana.edu.mx; Tel.: +52-(664)-607-8400 (ext. 234)

**Abstract:** This work presents a unique genetic programming (GP) approach that integrates a numerical local search method and a bloat-control mechanism within a distributed model for evolutionary algorithms known as EvoSpace. The first two elements provide a directed search operator and a way to control the growth of evolved models, while the latter is meant to exploit distributed and cloud-based computing architectures. EvoSpace is a Pool-based Evolutionary Algorithm, and this work is the first time that such a computing model has been used to perform a GP-based search. The proposal was extensively evaluated using real-world problems from diverse domains, and the behavior of the search was analyzed from several different perspectives. The results show that the proposed approach compares favorably with a standard approach, identifying promising aspects and limitations of this initial hybrid system.

**Keywords:** Genetic Programming; Bloat; NEAT; Local Search; EvoSpace

## 1. Introduction

Genetic programming (GP) is a branch of Evolutionary Computation (EC) dedicated to the artificial evolution of computer programs. In particular, even one of the earliest versions of GP, proposed by Koza in the 1990s and commonly referred to as tree-based GP or standard GP [1], continues producing strong results in difficult domains for more than 20 years later. In particular, we focus on an approach named neat-GP-LS [2] that integrates what we consider as fundamental elements of any state-of-the-art GP method, e.g., bloat control and local search (LS) techniques.

However, one discouraging aspect of integrating LS methods into a GP search is the increase in algorithm complexity (execution time might increase if the total number of generations is kept constant, but, since the algorithm converges more quickly, fewer generations are required to reach the same level of performance). One way to minimize this issue is by porting the search process to massively parallel architectures [3]. However, another approach is to move towards distributed EC systems (dEC) [4–6]. There are several possible benefits from this approach. First, it is much simpler to develop and use a distributed system than developing low-level code for GPUs or FPGAs [3,7]. The need for strict synchronization policies, for instance, is greatly reduced in a distributed framework compared to a GPU or FPGA implementation. Second, it is possible to leverage cheaper computing power that is already accessible, rather than investing in specialized hardware [8,9]. Finally, the robustness and asynchronous nature of an evolutionary search can easily deal with unexpected errors or dropped connections in a distributed environment. In this work, we use a distributed platform designed to run

using heterogeneous computing resources called EvoSpace, a conceptual model for the development of distributed pool-based algorithms [8–10]. While it has been applied in standard black-box optimization benchmarks and collaborative-interactive evolutionary algorithms [11], it has not been studied in a GP-based search.

In summary, this work presents a hybrid distributed GP system that integrates a recent and simple bloat control mechanism and a LS operator for parameter optimization of GP trees. For bloat control, neat-GP is used, a strategy that relies on speciation and fitness sharing to control code growth [12]. The LS approach is based on [13,14], where GP trees are augmented with weights for each node, which are then optimized with a trust region optimizer [15], a strategy that has also been applied to several real-world tasks [16,17]. This work shows that the EvoSpace model can easily exploit the speciation process performed by neat-GP, maintaining the same level of performance as the sequential version even though evolution is now performed in an asynchronous manner.

The remainder of this work is organized as follows. Section 2 presents relevant background and related research. Section 3 describes how the proposed system is ported to a distributed framework. A summary and conclusions are outlined in Section 4.

## 2. Background

This section described the neat-GP algorithm and a method to integrate LS in GP. In addition, a brief overview of EvoSpace model is provided.

### 2.1. neat-GP

The neat-GP algorithm [12] is based on the operator equalization [18] family of bloat control methods, in particular the Flat-OE [19] algorithms and the NeuroEvolution of Augmenting Topologies algorithm (NEAT) [20].

The main features of neat-GP are the following: The initial population only contains shallow trees (3 levels), while most GP algorithms initialize the search with small- and medium-sized trees (depth of 3–6 levels).

In neat-GP, individuals are grouped together into species (the terms *species* and *speciation* are used in a technical sense, following their use in NEAT, and are not meant to imply that they closely resemble their biological namesakes) based on their size and shape. With the following measure we can group individuals: given a tree $T$, let $n_T$ represent the size of the tree (number of nodes) and $d_T$ its depth (number of levels). Moreover, let $S_{i,j}$ represent the shared structure between both trees starting from the root node (upper region of the trees), which is also a tree, as seen in Figure 1. Then, the dissimilarity between two trees $T_i$ and $T_j$ is given by

$$\delta_T\left(T_i, T_j\right) = \beta \frac{N_{i,j} - 2n_{s_{i,j}}}{N_{i,j} - 2} + (1 - \beta) \frac{D_{i,j} - 2d_{s_{i,j}}}{D_{i,j} - 2}, \tag{1}$$

where $N_{i,j} = n_{T_i} + n_{T_j}$, $D_{i,j} = d_{T_i} + d_{T_j}$, and $\beta \in [0, 1]$; a degenerate case arises when both trees have a single node (only the root node), in this case $\delta_T = 0$.

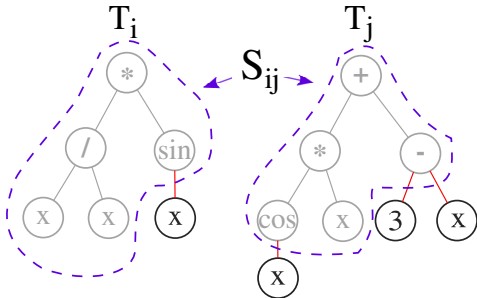

**Figure 1.** Example of the shared structure $S_{i,j}$ between two trees $T_i$ and $T_j$ ([2], with permission from Springer).

When a new individual $T_i$ is generated, it is compared with a single random individual $T_j$ from each species in sequence. Starting with a random ordering of the species, if $\delta_T(T_i, T_j) < h$, with threshold $h$ an algorithm parameter, then $T_i$ is assigned to the species to which $T_j$ belongs and no more comparisons are done. If the previous conditions is never met, then a new species is created for $T_i$.

Fitness sharing is used in each species, such that individuals that belong to large species are penalized while individuals that belong to smaller ones are less so. For a minimization problem, neat-GP uses a simple penalization given by

$$f'(T_i) = |S_u| f(T_i), \tag{2}$$

where $f(T_i)$ is the raw fitness of the tree, $f'(T_i)$ is the penalized or adjusted fitness, $S_u$ is the species to which $T_i$ belongs, and $|S_u|$ is the number of individuals in species $S_u$. The best individual of each species will be an exception; the fitness of this solution will not be penalized to preserve elite solutions within the population. The penalization only becomes important in the parent selection step, which is done based on the adjusted fitness, and it is done deterministically by sorting the population based on the adjusted fitness, making the process more elitist than traditional GP. Thus, highly penalized individuals might not produce offspring. However, if the best solution in a species is good enough, it can still survive and produce offspring since it is not affected by the penalization process.

### 2.2. Local Search in Genetic Programming

Particularly, we focus on symbolic regression problems, where the goal is to search for the symbolic expression $K^O : \mathbb{R}^p \to \mathbb{R}$ that best fits a particular training set $\mathbb{T} = \{(\mathbf{x}_1, y_1), \dots, (\mathbf{x}_n, y_n)\}$ of $n$ input/output pairs with $\mathbf{x}_i \in \mathbb{R}^p$ and $y_i \in \mathbb{R}$ defined as

$$(K^O, \boldsymbol{\theta}^O) \leftarrow \underset{K \in \mathbb{G}; \boldsymbol{\theta} \in \mathbb{R}^m}{arg\ min}\ f(K(\mathbf{x}_i, \boldsymbol{\theta}), y_i)\ with\ i = 1, \dots, n\ , \tag{3}$$

where $\mathbb{G}$ is the solution or syntactic space defined by the primitive set $\mathbb{P}$ of functions and terminals; $f$ is the fitness function that is based on the difference between a program's output $K(\mathbf{x}_i, \boldsymbol{\theta})$ and the desired output $y_i$; and $\theta$ is a particular parametrization of the symbolic expression $K$, assuming $m$ real-valued parameters. The goal of the LS method is to optimize the parameters of each GP solution.

As described in [13,14], an additional search operator is integrated into the search process, a local optimizer that tunes the implicit parameters of GP trees. In this way, subtree mutation and crossover are used to explore syntax space while LS is used to optimize the evolved solutions in parameter space.

As suggested in [21], for each individual $K$ in the population, we add a small linear upper tree above the root node, such that $K' = \theta_2 + \theta_1(K)$ where $K'$ represents the new program output, while $\theta_1$ and $\theta_2$ are the first two parameters from $\theta$, as shown in Figure 2.

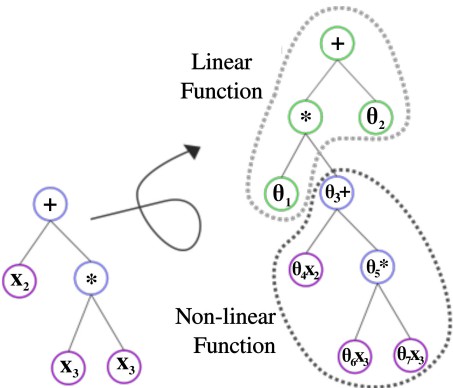

**Figure 2.** Example of the tree transformation for the LS process ([2], with permission from Springer).

In this way, for all the other nodes $n_k$ in the tree $K$ we add a weight coefficient $\theta_k \in \mathbb{R}$, such that each node is now defined by $n'_k = \theta_k n_k$, where $n'_k$ is the new modified node, $k \in \{1, ..., r\}, r = |Q|$ and $Q$ is the tree representation. Notice that each node has a unique parameter that can be modified to help meet the overall optimization criteria of the non-linear expression. At the beginning of the GP run each parameter is initialized by $\theta_k = 1$. During the GP syntax search, subtrees belonging to different individuals are swapped, added or removed (following the standard crossover/mutation rules) together with its corresponding parameters, without affecting their values. Therefore, we consider each tree as nonlinear expression and the local search operator must now find the best fit parameters of the model $K'$. The problem can be solved using a variety of techniques, but, following the authors [13,14], we employ the trust region algorithm [22].

One of the most important things to consider is that the local search optimizer can substantially increase the underlying computational cost of the search, particularly when individual trees are very large. While applying the local search strategy to all trees might produce good results [13], it is preferable to reduce to a minimum the amount of trees to which it is applied.

## 2.3. Integration LS into neat-GP

The neat-GP-LS algorithm [2] is a proposal of the integration of neat-GP search with LS process. Figure 3 represents the main stages in the neat-GP-LS algorithm process. The algorithm reveals that combining both methods improves the quality of the solutions found, and allows the search process to generate very compact solutions. In addition, it demonstrates that the variance in the search is quite small and the size of the solutions are problem dependent, which means the search tends to converge toward solutions that lie in the same regions of solution space, i.e. in the same species.

This entails that one important aspect of the neat-GP search and hence of neat-GP-LS is the use of speciation to generate a diverse population of individuals, in terms of size and shape, demonstrating four relevant information about speciation.

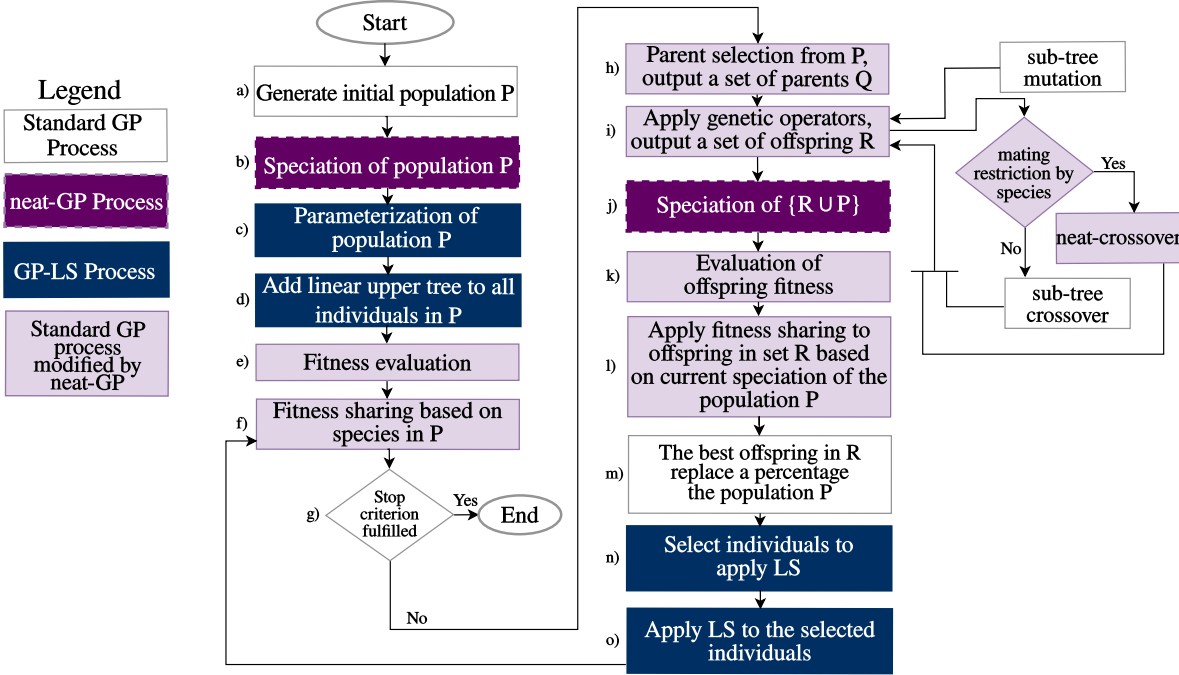

**Figure 3.** General flow diagram of the neat-GP-LS algorithm ([2], with permission from Springer).

First, it produces species that grow in size when they contain highly fit individuals, particularly when they contain the best individual in the population. Second, the size and depth of the models grow over time, but diversity is maintained with different species in the population. Third, while solutions are diverse in their structure, many species tend to contain at least one highly fit

model. Finally, the algorithm shows that it allows a species to grow in size, when it contains the best individual; this increased exploitation seems to be justified since the LS operator tends to produce the highest levels of improvements in that particular species.

### 2.4. EvoSpace

The EvoSpace model for evolutionary algorithms (EA) follows a pool-based approach [8,9], where the search process is conducted by a collection of possibly heterogeneous processes that cooperate using a shared memory or population pool. We refer to such algorithms as pool-based EAs (PEAs) and highlight the fact that such systems are intrinsically parallel, distributed and asynchronous.

In EvoSpace, distributed nodes (called EvoWorkers) asynchronously interact with the pool; their job is to take a subset of individuals from the central pool, which is called a sample, and evolve them for a certain number of generations (or until a given termination criterion is met), and return the new population of offspring back to the pool. The general scheme is depicted in Figure 4.

This means that EvoSpace has two main components, a set of EvoWorkers and a single instance of an EvoStore. The EvoStore container manages a set of objects representing individuals in a EA population. EvoWorkers pull a subset of individuals from the EvoStore making them unavailable to other workers. Moreover, individuals are removed from the EvoStore as a random subset or sample of the population. Once a EvoWorker has a sample to work on, it can perform a partial evolutionary process, and then return the newly evolved subpopulation to the EvoStore where the new individuals replace those found in the original sample; at this point, replaced or reinserted individuals can be taken by others clients. Figure 5 shows the distributed architecture of the EvoSpace model with GP. The figure shows that on the Server the EvoSpace manager and HTTP communication framework are performed, while different samples of individuals from the population are sent to EvoWorkers where evolution takes place.

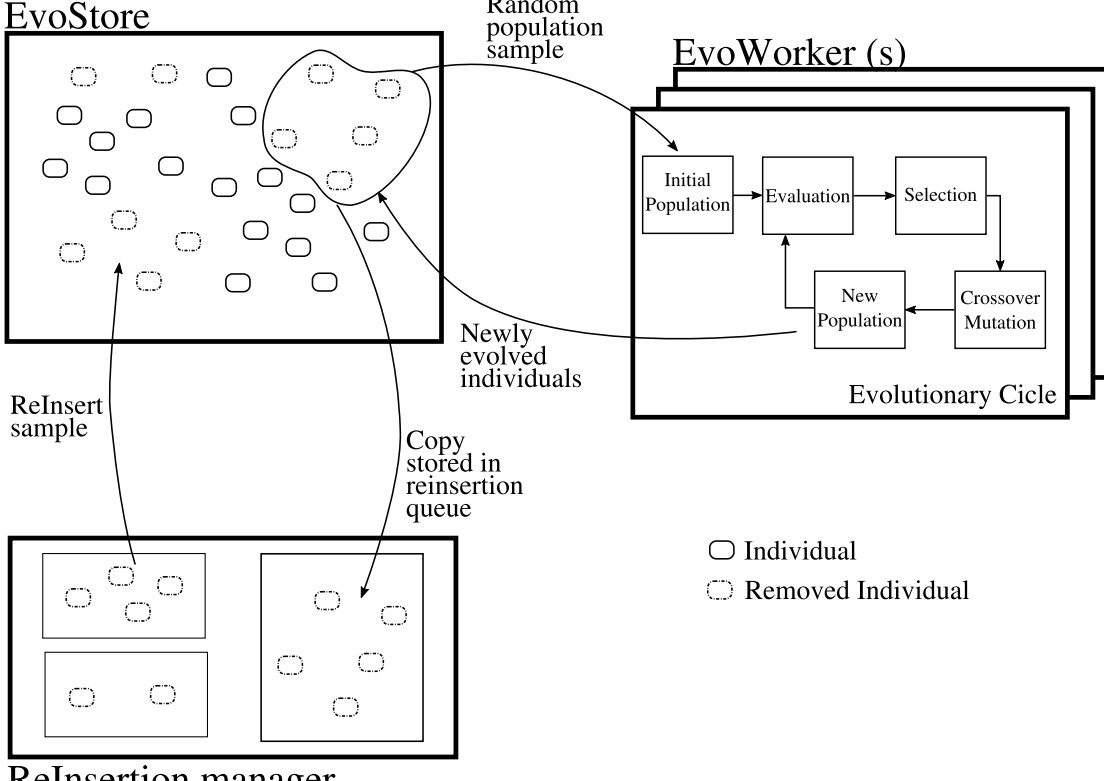

**Figure 4.** Main components and data flow within the EvoSpace model.

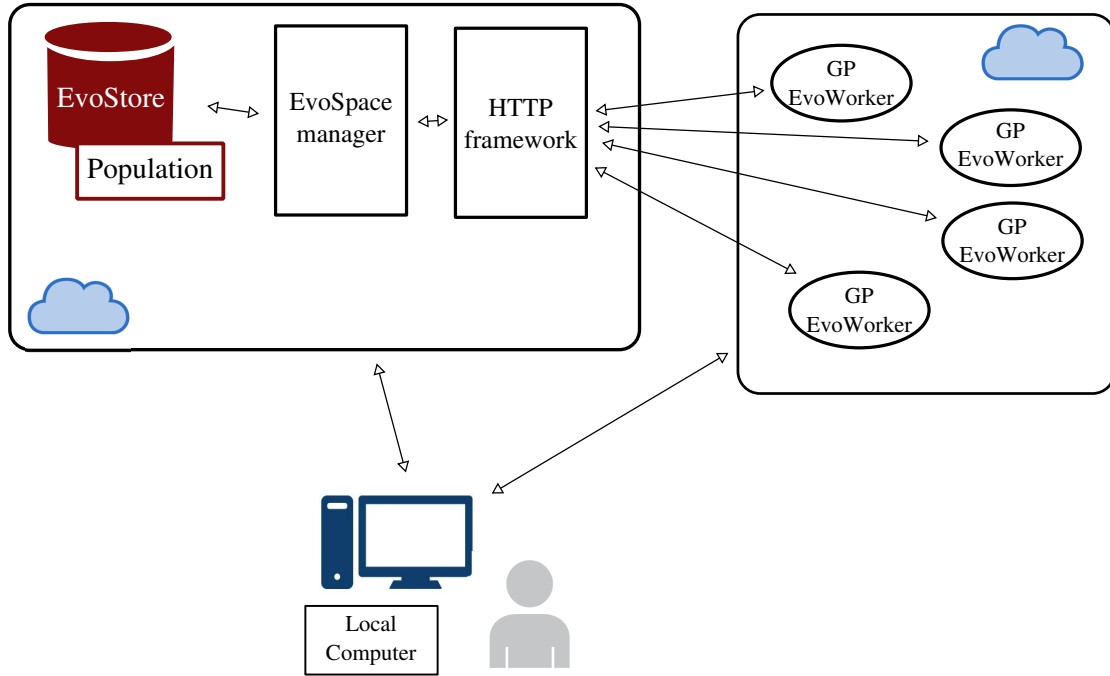

**Figure 5.** EvoSpace distributed architecture.

EvoSpace was conceived as a model for cloud-based evolutionary algorithms and is general enough to be amenable to any type of population-based algorithm. Several works have shown that this general approach can solve standard black-box optimization problems [9] and even interactive evolution tasks [11]. It has been shown, as expected, that distributing costly fitness function evaluations will help reduce the total run-time of the algorithm [9].

## 3. Distributing neat-GP-LS into the EvoSpace Model

In this work, we present the first implementation of a GP algorithm on EvoSpace.

Since neat-GP-LS already divides the population into species, it seems straightforward to exploit this structure and distribute individuals to EvoWorkers by sending complete species to each.

### 3.1. The Intra-Species Distance and Re-Speciation

One aspect of neat-GP-LS that is not asynchronous is the speciation process. In the sequential and synchronous versions, speciation occurs at specific moments during the search, as shown in Figure 3. However, since EvoSpace is asynchronous, EvoWorkers return samples to the population pool at different moments in time. When an EvoWorker returns a sample, it is not correct to assume that all of the new individuals actually belong in the same species. It is possible that the species diverged during the local evolution carried out on the EvoWorker.

To solve this issue, we track the level of homogeneity within each species, which is measured before a species leaves the pool and when the new species returns from the EvoWorker. If a significant change is detected, then a flag is raised that tells EvoSpace that the population should go through a new speciation process or re-speciation. This is done by computing what is referred to as the intra-species distance. Basically, in each species, we compute the dissimilarity measure using Equation (1), between each tree $T_i$ and its nearest neighbor $T_j$ (the individual with which Equation (1) is minimum within the species), calling this value $nn_{T_i}$. Then, the intra-species distance $D_{S_l}$ for species $S_l$ is the average of all $nn_{T_i}$ considering all $T_i \in S$.

The $D_{S_l}$ values could be used in different ways to trigger a re-speciation process. In this work, we can say that $D_{S_l}$ is the intra-species distance before $S_l$ is taken as a sample by an EvoWorker, and we can define $\hat{D}_{S_l}$ as the intra-species distance of species $S_l$ computed with the population returned

by the EvoWorker. If $\hat{D}_{S_l} > D_{S_l}$ for any species in the population, then a re-speciation event is triggered. Basically, this causes a synchronization event, where the EvoStore waits for all species to return and the population goes through the speciation process once more. Figure 6 shows the basic scheme of the proposes implementation. Compared to Figure 5, the new implementation in Figure 6 accounts for specific elements of the neat-GP algorithm. In particular, the speciation process is carried out on the server, such that instead of sending random samples of individuals to the EvoWorkers, complete species are sent and a local evolutionary process is carried out. In this case, the number of EvoWorkers used depends on the number of species in the population.

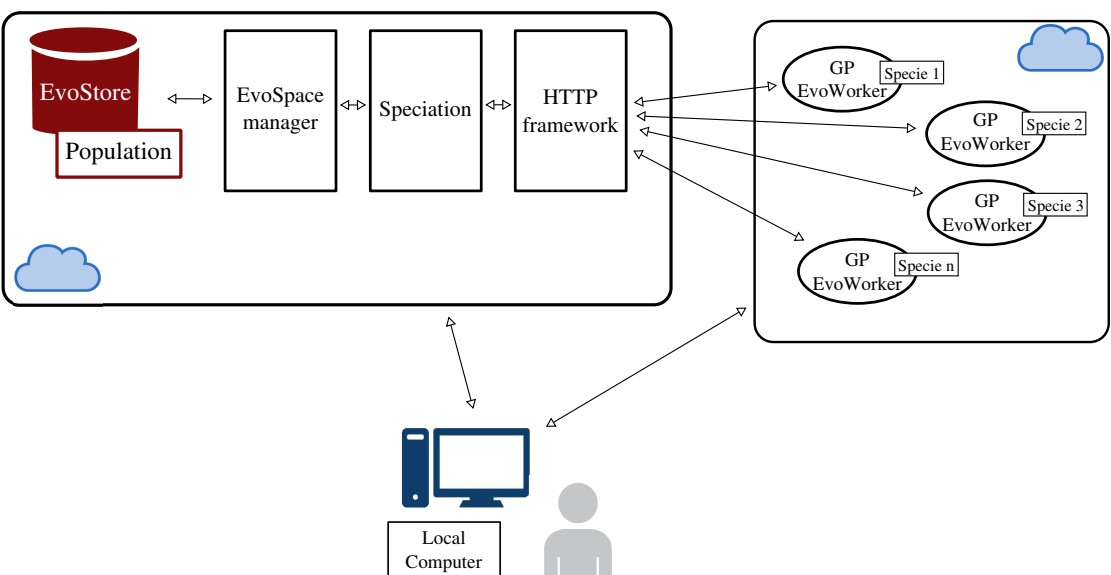

**Figure 6.** Implementation of the neat-GP-LS algorithm in EvoSpace, where the samples taken by each EvoWorker correspond to a complete species.

### 3.2. Experiments and Results

We analyzed and evaluated the integration of the neat-GP-LS algorithm in a PEA known as the EvoSpace model. EvoSpace was designed for problems where fitness computation might be expensive; in this work, we were only interested in studying the effects of implementing neat-GP-LS as a PEA. In particular, we wanted to determine if there are any significant and substantial effects on the convergence of the algorithm, the solutions qualities on all the population and the behavior of the bloating phenomena.

For simplicity, the distributed framework was simulated using multiple CPU threads, such that each EvoWorker was assigned to a specific thread. When the number of EvoWorkers exceeded the number of threads, then several workers could share a single thread.

All experiments were carried out using real world symbolic regression problems, where the objective is to minimize the fitness function. All problems are summarized in Table 1.

When a species was sent to an EvoWorker, we performed a short local evolutionary search, basically a standard GP search using the parameters specified in Table 2. The number of EvoWorkers depended on the number of species in the EvoStore, and we assumed that an EvoWorker was always available for any species in the EvoStore. In addition, the local evolution performed in an EvoWorker iterated for 10 generations, applying the LS operator with probability of 0.50.

**Table 1.** Symbolic regression real world problems.

| Problems | No. Instances | No. Features | Description |
|---|---|---|---|
| *Housing* [23] | 506 | 14 | Concerns housing values in suburbs of Boston. |
| *Concrete* [24] | 1030 | 9 | The concrete compressive strength is a highly nonlinear function of age and ingredients. |
| *Energy Heating* [25] | 768 | 9 | This study looked into assessing the heating load requirements of buildings as a function of building parameters. |
| *Energy Cooling* [25] | 768 | 9 | This study looked into assessing the cooling load requirements of buildings as a function of building parameters. |
| *Tower* [26] | 5000 | 26 | An industrial data set of a gas chromatography measurement of the composition of a distillation tower. |
| *Yacht* [27] | 308 | 7 | Delft data set, used to predict the hydodynamic performance of sailing yachts from dimensions and velocity. |

**Table 2.** Parameters used in real world problems.

| Parameter | neat-GP-LS |
|---|---|
| Runs | 30 |
| Population | 100 |
| Generations | 10 |
| Training set | 70% |
| Testing set | 30% |
| Operators Crossover ($p_c$), Mutation ($p_m$) | $p_c$=0.9, $p_m$=0.1 |
| Tree initialization | Ramped Half-and-Half, maximum depth 6. |
| Function set | +,-,x,sin,cos,log,sqrt,tan,tanh, constants |
| Terminal set | Input variables and constants as indicated in each real-world problem. |
| Selection for reproduction | Eliminate the $p_{worst} = 50\%$ worst individuals of each species. |
| Elitism | Do not penalize the best individual of each species. |
| Species threshold value | $h = 0.15$ with $\beta = 0.5$ |
| Local optimization probability | $P_s = 0.5$ |

Figure 7 shows a single run of the PEA version of neat-GP-LS on the Housing, Concrete and Energy Cooling problems. The plots show the convergence of the training and testing RMSE, as well as the average size of the population given in number of tree nodes. The horizontal axis represents the number of samples taken from the EvoStore. Note that the number of samples over different problems and over different runs l varied due to the randomness of the individual population and the speciation process, and due to the asynchronous nature of the EvoSpace model, which makes it unfeasible to aggregate the behavior of multiple runs into a single plot. Therefore, these plots only show a single run, but the behavior of the algorithm in these examples is in fact representative of the convergence behavior of most runs. One notable observation is the almost identical behavior of both training and testing MAE in all of the runs, showing that the algorithm generalizes in a consistent manner relative to training performance. The size of the population is also quite informative. Notice that, while the average size fluctuates in all cases, the algorithm is in general producing compact solutions. This is particularly clear when the search process terminates and the final sample is returned to the EvoStore.

The results are summarized in Figures 8 and 9, which show a box plot comparisons between the sequential neat-GP-LS algorithm and the PEA implementation in EvoSpace, respectively, for test RMSE and the average size of the population. Table 3 presents the p-values of the Friedman test, where bold values indicate that the null hypothesis is rejected at the $\alpha = 0.05$ confidence level. The null hypothesis

states that the medians of the two groups are the same. Notice that, on three (Concrete, Energy Heating and Tower) out of the six problems, the EvoSpace version performed worse than the sequential algorithm in terms of RMSE, since the null-hypothesis were rejected. Conversely, if we consider the three problems in which the PEA version and the sequential algorithm performed equivalently based on test RMSE (i.e., the null hypothesis is not rejected), the Housing Energy Cooling and Yacht problems, EvoSpace produced smaller trees and thus was more effective at bloat control. Therefore, we can state with some confidence that the modified search dynamics introduced in the distributed version of the algorithm do alter the effectiveness of the search. On the one hand, the quality of the results seemed to depend on the problem. On the other hand, in all cases where the EvoSpace implementation achieved equivalent performance, it was significantly and substantially less affected by bloat, producing more parsimonious and compact solutions.

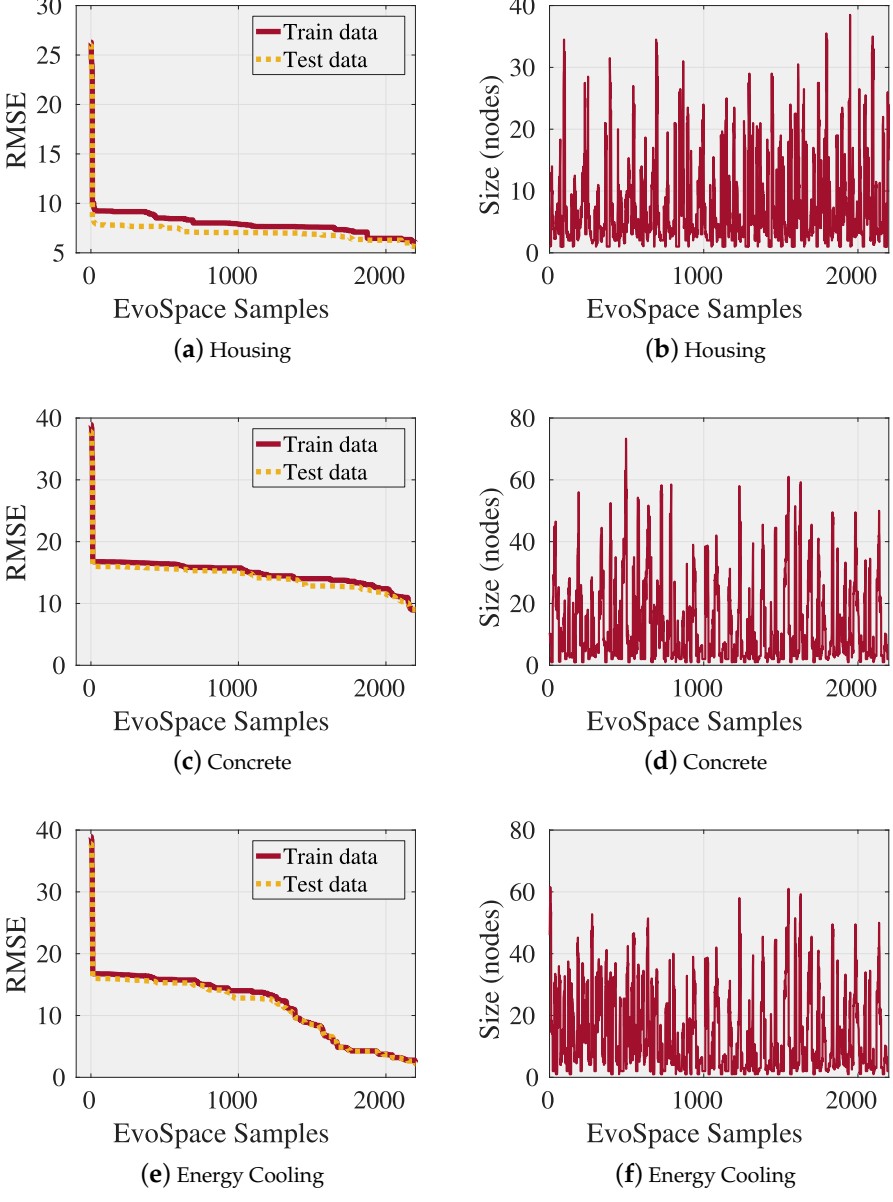

**Figure 7.** Performance of a single run of the PEA implementation of neat-GP-LS in EvoSpace for: *Housing* (**a**), (**d**); *Concrete* (**b**), (**e**); and *Energy Cooling* (**d**), (**f**). The plots in the left column show the evolution of the training and testing RMSE. The plots in the right column show the evolution of the average program size. All plots are ordered based on the number of samples taken from the EvoStore.

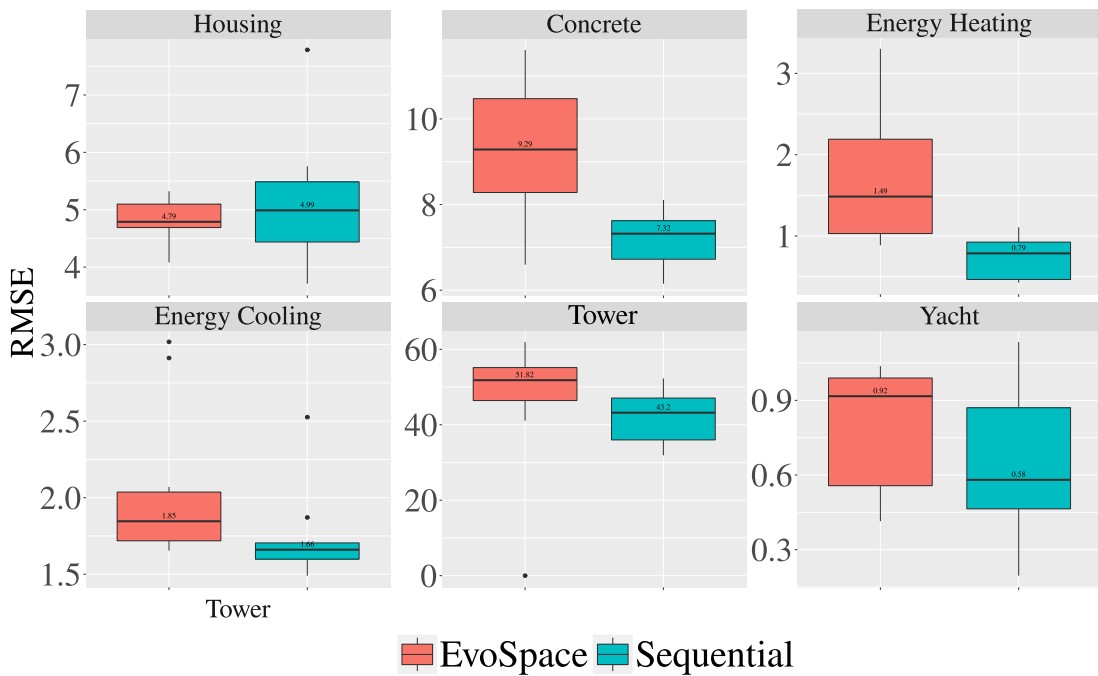

**Figure 8.** Box plot comparison of the sequential and the EvoSpace implementation of the neat-GP-LS algorithm on the testing RMSE.

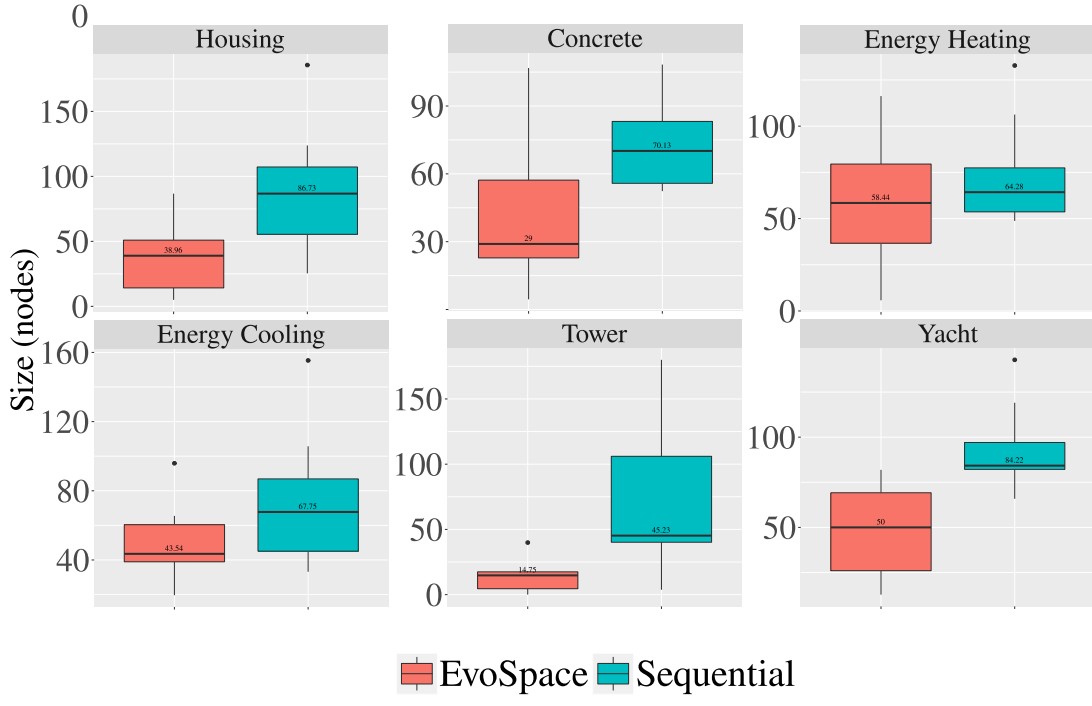

**Figure 9.** Box plot comparison of the sequential and the EvoSpace implementation of the neat-GP-LS algorithm on the average size of individuals given in number of nodes.

It is reasonable to assume that larger learning problems, in terms of number of instances and features, are in general more difficult to solve. Moreover, difficult problems usually require more complex or larger solutions to effectively model their structure. The three problems where RMSE performance of the EvoSpace implementation was statistically worse (Concrete, Energy Heating and Tower) are also three of the four largest problems used in our experiments, in terms of total number

of instances and number of features (see Table 1). Since the EvoSpace search dynamics pushes the search towards smaller program sizes, with statistical significance in five of the six problems (including all problems in which RMSE performance was worse), a plausible explanation of the results can be formulated. The EvoSpace implementation is controlling bloat too aggressively, severely impacting learning in the more difficult test cases. Therefore, future variants of the implementation will need to allow the search to explore large program sizes to evolve more accurate models.

**Table 3.** Friedman test *p*-values, comparing the sequential neat-GP-LS and the EvoSpace implementation based on test RMSE and average size of the final population. Bold indicates that the null-hypothesis was rejected at the $\alpha = 0.05$ significance level.

|                 | test   | size   |
|-----------------|--------|--------|
| **Problem**     | \multicolumn{2}{c}{*p*-value} |        |
| Housing         | 0.2733 | **0.0114** |
| Concrete        | **0.0010** | **0.0010** |
| Energy Cooling  | 0.0578 | **0.0285** |
| Energy Heating  | **0.0114** | 1.000  |
| Tower           | **0.0285** | **0.0114** |
| Yacht           | 0.2059 | **0.0114** |

Finally, Figure 10 analyzes the re-speciation process based on the intra-species distance. The plot shows how $D_{S_l}$ changes over for each of the species in the population, using a single run of the algorithm on the Housing problem, zooming in on the first 225 samples taken by the EvoWorkers. Each vertical line represents the difference between $D_{S_l}$ and $\hat{D}_{S_l}$. When a line is black (shorter lines), it means that a re-speciation event was not triggered, and when a line is red (longer lines) this means that a re-speciation event could have been triggered by a sample. We can see that, at the beginning of the run, speciation events are more frequent and, as the search progresses begins to converge, these events become infrequent.

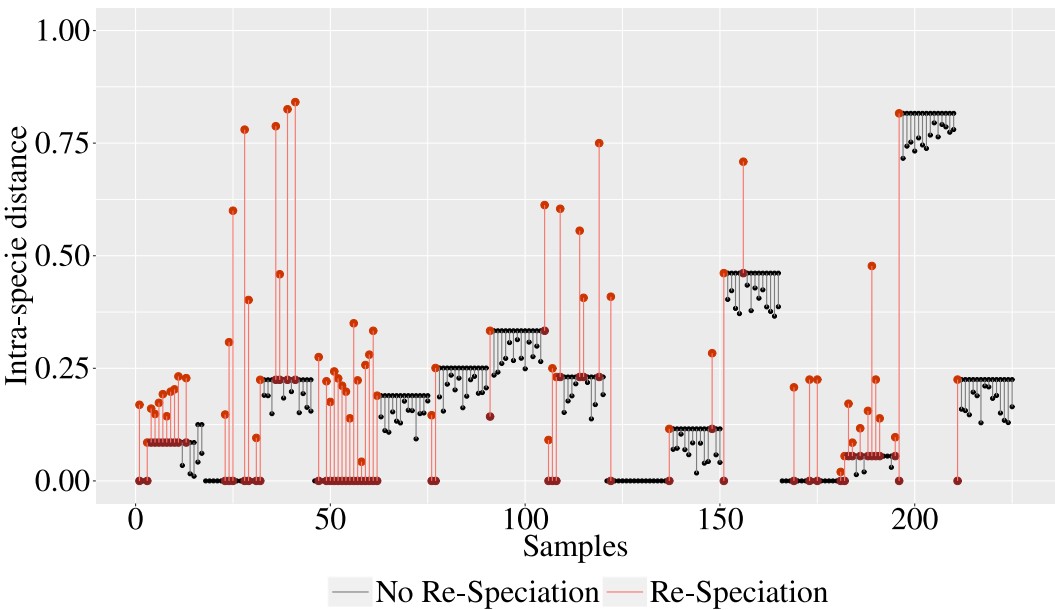

**Figure 10.** Analysis of the re-speciation process using the intra-species distance.

## 4. Conclusions and Future Work

This work presents, to the authors' knowledge, the first implementation of a GP system in a Pool-based EA, using the EvoSpace model. The PEA approach is particularly well suited for the speciation-based neat-GP search, allowing for a straightforward strategy to distribute the population over the processing elements of the system (EvoWorkers). It is notable that the performance of the PEA version was not equivalent to the sequential one, in two key respects. On the one hand, it did not reach the same level of performance on some problems. On the other hand, on the problems where it performed equivalently, or better, it was able to reduce solution size significantly.

Future work will center around eliminating the synchronization required by the speciation process in the EvoSpace implementation. Another interesting extension is to consider other elements in the speciation process besides program size and shape, such as program semantics, program behavior or solution novelty. Moreover, we would like to integrate a wider range of parameter local search methods, particularly gradient free methods, and to combine them with other forms of local optimizers that work at the level of syntax or semantics. Finally, it will be important to deploy the proposed algorithms in high-performance computing platforms, to tackle large scale big data problems, where distributing the computational load becomes a requirement.

**Author Contributions:** L.T., P.J.-S. and M.G.-V. conceived and designed the experiments; P.J.-S. performed the experiments; P.J.-S., L.T., F.F.d.V. and F.C. analyzed the data; F.F.d.V. and F.C. contributed analysis tools; P.J.-S. and L.T. wrote the paper; and M.G.-V., F.F.d.V. and F.C. provided feedback and improved the manuscript.

**Acknowledgments:** This work was funded by CONACYT (Mexico) project No. FC-2015-2/944 Aprendizaje evolutivo a gran escala, and TecNM (Mexico) project no. 6826-18-p. The first author was supported by CONACYT doctoral scholarship 332554. The authors would like to thank Spanish Ministry of Economy, Industry and Competitiveness and European Regional Development Fund (FEDER) under projects TIN2014-56494-C4-4-P (Ephemec) and TIN2017-85727-C4-4-P (DeepBio); and Junta de Extremadura Project IB16035 Regional Government of Extremadura, Consejeria of Economy and Infrastructure, FEDER.

**Conflicts of Interest:** The authors declare no conflict of interest.

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
