# Peer review of "Pool-Based Genetic Programming Using Evospace, Local Search and Bloat Control"

_mca, doi:10.3390/mca24030078_

Round 1

Reviewer 1 Report

This paper proposes a hybrid distributed GP system that integrates two techniques one to control bloat and the other to optimize constants, i.e., to perform a local search. 

The manuscript is easy to read and follow, and the idea is well-presented. 

Regarding the fitness sharing strategy, I assume that the process is to minimize the fitness function; this fact should be mentioned to understand the description appropriately. 

Figure 5 and 6 could be explained in the description of each algorithm; for instance, the last paragraph of Section 2 for the former. 

The propose is to implement GP in a distributed environment; however, I could not see any information regarding the time, this is interesting in the setting used because the communication time is insignificant given that you are using threads. 

Author Response

This paper proposes a hybrid distributed GP system that integrates two techniques one to control bloat and the other to optimize constants, i.e., to perform a local search.

RESPONSE: We thank the reviewer for her(his) time and effort reviewing our manuscript.

The manuscript is easy to read and follow, and the idea is well-presented.

RESPONSE: We thank the reviewer for the kind comment.

Regarding the fitness sharing strategy, I assume that the process is to minimize the fitness function; this fact should be mentioned to understand the description appropriately.

RESPONSE: This comment is present in final paragraph of page 2, where it is said that "For a minimization problem ...".

Figure 5 and 6 could be explained in the description of each algorithm; for instance, the last paragraph of Section 2 for the former.

RESPONSE: For Figure 5 it was added to the end of the first paragraph of page 5. For Figure 6 it is added at the end of section 3.1, second paragraph of page 6.

The propose is to implement GP in a distributed environment; however, I could not see any information regarding the time, this is interesting in the setting used because the communication time is insignificant given that you are using threads.

RESPONSE: Evaluation based on run time will be carried out in a future work, using a cloud-based distributed system, which is the final goal of this research. What can be said for now is that time will be dependent on several factors, including bandwith, population size, dataset size and complexity of the function set; all of these expects are to be studied in future work.

Reviewer 2 Report

This work proposed a NEAT GP system with linear scaling which is distributed in the EvoSpace model. The idea is interesting. The paper is well organised and well written. The experiment design is reasonable and the encouraging results are shown clearly. In general, this is a good work. However, the paper can be improved in different aspects.

Detail comments:
1). What is the major motivation for using EvoSpace? It is not very clear why EvoSpace. Is there any other distributing technique? Please provide more justifications.

2). Compared with previous work on GP [20], [13], [14], the major contribution of work is using EvoSpace. You need to make this point very clear.

3). A deep analysis of the result is expected while now most of the section only presents and compares the results. Please try to analysis why on some dataset the proposed system could not work well while on other datasets it can have good performance.

4) Fig.7 is only based on one run. How to choose this one run? The pattern on one run could not represent all the runs, which makes the pattern not very convincing.

Author Response

This work proposed a NEAT GP system with linear scaling which is distributed in the EvoSpace model. The idea is interesting. The paper is well organised and well written. The experiment design is reasonable and the encouraging results are shown clearly.

RESPONSE: We thank the reviewer for her (his) time and work put into the review of our manuscript.

In general, this is a good work. However, the paper can be improved in different aspects.

RESPONSE: We did our best to make the suggested improvements.

Detail comments:
1). What is the major motivation for using EvoSpace? It is not very clear why EvoSpace. Is there any other distributing technique? Please provide more justifications.

RESPONSE: Our motivation was indirectly mentioned in page 2, second paragraph, where we wrote that "the EvoSpace model can easily exploit the speciation process performed by neat-GP", which was the main factor in choosing EvoSpace over other distributed, or more specifically other pool-based, evolutionary algorithms. This is now more clearly stated in this same paragraph.

2). Compared with previous work on GP [20], [13], [14], the major contribution of work is using EvoSpace. You need to make this point very clear.

RESPONSE: We agree with the reviewer. In fact, this is only indirectly referenced in the same paragraph as mentioned in the previous point. Therefore, this entire paragraph (next to last in the introduction), has been re-written to highlight both the main contribution and motivation behind the work.

3). A deep analysis of the result is expected while now most of the section only presents and compares the results. Please try to analysis why on some dataset the proposed system could not work well while on other datasets it can have good performance.

RESPONSE: We have added a discussion paragraph starting at the last paragraph of page, where our analysis of the results and observed behavior is given.

4) Fig.7 is only based on one run. How to choose this one run? The pattern on one run could not represent all the runs, which makes the pattern not very convincing.

RESPONSE: Indeed, the comment by the reviewer is pertinent, but we justify it on the following grounds. First, as is now stated in the manuscript

"... the number of samples over different problems, and over different runs, will vary due to the randomness of the individual population and the speciation process."

This makes it unfeasible to average over multiple runs, since the algorithm is asynchronous. However, we can state, that the behavior of the plots is representative of all runs performed in out experiments, as is stated now clearly in the text.